# Antimicrobial Susceptibility of *Campylobacter* spp. Isolated from Cattle in Mongolia

**DOI:** 10.3390/vetsci12100931

**Published:** 2025-09-24

**Authors:** Erdenebat Bulgan, Zolzaya Byambajav, Batsukh Naranchimeg, Batsaikhan Chantsal, Tsognemekh Bolormaa, Badrakh Sandagdorj, Purevdorj Nyam-Osor, Eisaku Kikuchi, Akio Suzuki, Jirachaya Toyting-Hiraishi, Toyotaka Sato, Motohiro Horiuchi

**Affiliations:** 1Laboratory of Veterinary Hygiene, Faculty of Veterinary Medicine, Graduate School of Infectious Diseases, Hokkaido University, Sapporo 060-0818, Japan; erdenebat.bulgan.p8@elms.hokudai.ac.jp (E.B.); susan@vetmed.hokudai.ac.jp (A.S.); j-toyting@vetmed.hokudai.ac.jp (J.T.-H.); sato.t@vetmed.hokudai.ac.jp (T.S.); 2Laboratory of Veterinary Sanitation and Hygiene, Ulaanbaatar Veterinary Department, Ulaanbaatar 16050, Mongolia; zoloo_wow20@yahoo.com (Z.B.); bsunshine1984@gmail.com (B.N.); 3Department of Veterinary Public Health, School of Veterinary Medicine, Mongolian University of Life Sciences, Ulaanbaatar 17024, Mongolia; chantsal@muls.edu.mn (B.C.); ts.bolormaa@muls.edu.mn (T.B.); sandagdorj_vet@muls.edu.mn (B.S.); nyam-osor@muls.edu.mn (P.N.-O.); 4One Health Research Center, Hokkaido University, Sapporo 060-0818, Japan; kikuchi@ohrc.hokudai.ac.jp; 5Veterinary Research Unit, International Institute for Zoonosis Control, Hokkaido University, Sapporo 060-0818, Japan; 6Global Station for Zoonosis Control, Global Institute for Collaborative Research and Education, Hokkaido University, Sapporo 060-0818, Japan

**Keywords:** *Campylobacter*, antimicrobial resistance, fluoroquinolone, ciprofloxacin, cattle

## Abstract

This study aimed to assess prevalence and antimicrobial resistance of *Campylobacter* spp. in cattle in Mongolia. Thirty-five *Campylobacter* spp., including 23 *C. jejuni*, 7 *C. hyointestinalis*, 4 *C. fetus*, and 1 *C. lari*, were isolated. Multilocus sequence typing of *C. jejuni* cattle isolates revealed substantial genetic diversity. The antimicrobial resistance patterns of the *C. jejuni* cattle isolates completely differed from those of previously reported chicken isolates; excluding one ciprofloxacin-resistant isolate, all *C. jejuni* isolates were susceptible to tetracycline and ciprofloxacin. To the best of our knowledge, this is the first report on the characterization of *Campylobacter* spp. in cattle in Mongolia.

## 1. Introduction

*Campylobacter* infection is a leading cause of foodborne bacterial diseases worldwide. *Campylobacter* spp. colonize in the intestines of homoiothermal animals asymptomatically in most cases except for *C. fetus* infection in cattle, which causes bovine genital campylobacteriosis. Among the *Campylobacter* spp., *C. jejuni* and *C. coli* are the predominant causes of campylobacteriosis in humans. Additionally, species such as *C. fetus*, *C. lari*, and *C. hyointestinalis* are also known to be pathogenic to humans [1,2]. The primary clinical symptoms of *Campylobacter* infection are diarrhea, abdominal pain, fever, headache, nausea, and vomiting. Although most cases are self-limited within 3–6 days without any treatment, macrolides and fluoroquinolones (FQs) are prescribed to patients with severe or prolonged symptoms. *Campylobacter* infection is recognized as a risk factor for Guillain-Barré syndrome (GBS), an autoimmune demyelinating polyneuropathy [3]. Nearly 30% of GBS cases are associated with antecedent infection of *Campylobacter* [4,5], and about 0.1% of infected individuals develop GBS [6].

Livestock such as poultry and ruminants are major attributed sources of *Campylobacter* infection in humans, and companion animals and environments including surface water are also sources of infection [7,8]. As the route of infection can vary by region depending on differences in livestock breeding conditions, livestock hygiene management, food hygiene practice, food culture, and the coexisting environment between humans and animals, the identification of the source and infection route to humans is important for implementing effective measures against campylobacteriosis.

In Mongolia, meat is thoroughly cooked before consumption; thus, it is unlikely that undercooked meat poses a risk of infection. However, there are other risks of foodborne campylobacteriosis such as ingestion of traditional dairy products made from unpasteurized animal milk, secondary contamination during processing and along the food chain, the spread of ready-to-eat foods such as salads, the westernization of dietary habits, and the use of surface water in rural areas.

We recently reported the high antimicrobial resistance (AMR) ratios of *Campylobacter* spp. in chickens in Mongolia. All isolates were resistant to tetracycline (TET), and more than half were resistant to FQs. Half of the *C. coli* isolates exhibited muti-drug resistance, including resistance to erythromycin (EM) [9]. Because WHO assigned EM and ciprofloxacin (CPFX) as critically important antimicrobials for treating severe campylobacteriosis in humans, the presence of macrolide- and/or FQ-resistant *Campylobacter* spp. in food-producing animals, foods, and environments represents a public health concern. However, no data on the existence and AMR of *Campylobacter* spp. in other livestock and contamination of *Campylobacter* spp. in food chain are available. Additionally, campylobacteriosis in humans has not been officially reported to date, probably because of the insufficient diagnostic capacity.

More than 5 million cattle, 25 million sheep, and 23 million goats are raised in Mongolia. Meat is one of the primary food products in Mongolia, and beef and mutton are widely consumed. The annual consumption is approximately 134,000 tons for beef and 275,000 tons for sheep and goat meat [10]. Additionally, Mongolia has a unique culture of using animal milk and consumes 160,000 tons of milk per year. These facts imply that ruminants are possible sources of *Campylobacter* infection in humans. Therefore, in the current study, we characterized *Campylobacter* spp. isolated from cattle to provide scientific data for investigating the zoonotic transmission route of *Campylobacter* spp. in Mongolia.

## 2. Materials and Methods

### 2.1. Sampling

In total, 100 rectal swabs were collected from 10 dairy farms in Ulaanbaatar, Mongolia (Table 1). Most samples were collected from the Songinokhairkhan district, a cattle farm-concentrated area in Ulaanbaatar. Rectal swabs were collected using sterile cotton swabs. Each swab was directly placed in 5 mL of Brucella broth (Becton Dickinson, Franklin Lakes, NJ, USA) supplemented with Modified Preston Campylobacter Supplement (Oxoid, Basingstoke, UK) and 5% lysed horse blood (LHB, Nippon Bio-test Laboratories Inc., Asaka, Japan; the supplemented broth is hereafter termed Preston broth) and kept cooled during transportation to the laboratory.

### 2.2. Isolation and Identification of Campylobacter spp.

Preston broth containing the rectal swabs was incubated for 24 h at 42 °C under microaerobic conditions using AnaeroPacks-MicroAero (Mitsubishi Gas Chemical Co., Inc., Tokyo, Japan) to enrich thermophilic *Campylobacter* spp. Then, one loop of the broth was streaked on *Campylobacter* Blood-Free Selective Agar Base (Oxoid) with Charcoal Cefoperazone Deoxycholate Agar (CCDA) selective supplement (Oxoid; hereafter termed mCCDA), and the mCCDA plates were incubated for 48 h at 37 °C under microaerobic conditions. Typical *Campylobacter* colonies on the mCCDA plates were picked up using sterilized toothpicks and inoculated into *Campylobacter* Blood-Free Selective Agar Base without CCDA supplement to analyze bacterial growth under microaerobic and anaerobic conditions using AnaeroPacks-MicroAero and AneroPak-Anaero (Mitsubishi Gas Chemical Co., Inc.), respectively.

The suspected *Campylobacter* colonies were cultured in 2 mL of Brucella broth for 48 h at 37 °C under microaerobic conditions, and DNA was extracted from the cultures using CICA Geneus DNA Extraction Reagent (Kanto Chemical Co., Inc., Tokyo, Japan) according to the supplier’s instructions. Species-specific multiplex PCR targeting *cdtC* gene [11], which can distinguish *C. jejuni, C. coli,* and *C. fetus*, was performed as described previously [9]. *Campylobacter* spp. were also confirmed by matrix-assisted laser desorption/ionization (MALDI) time-of-flight mass spectrometry using the MALDI Biotyper Compass (Bruker Daltonics, Bremen, Germany).

### 2.3. Antimicrobial Susceptibility Test

The antimicrobial susceptibility of the isolates was analyzed using the E-test and broth dilution method. E-test strips of TET, EM, nalidixic acid (NA), and ciprofloxacin (CPFX) were purchased from BioMérieux (Marcy-l’Etoile, France), and E-test was performed as descrived previously [9]. The minimum inhibitory concentrations (MIC) for NA were also determined using the broth dilution method according to the CLSI guidelines. Antimicrobial susceptibility was determined according to CLSI M45 third edition, with resistant breakpoints of EM (≥32 μg/mL), CPFX (≥4 μg/mL), and TET (≥16 μg/mL), and according to Japanese Veterinary Antimicrobial Resistance Monitoring System (JVARM) for NA (≥16 μg/mL).

### 2.4. Genetic Analysis

Nucleotide sequencing of the *gyrA* gene to analyze codon 86 amino acid substitution for estimating FQ susceptibility and 16S rDNA for species identification, was performed as described elsewhere [9]. Multi-locus sequence typing (MLST) was performed to identify sequence types (STs) and clonal complexes (CCs) of the *C. jejuni* isolates. Seven housekeeping genes (*aspA*, *glnA*, *gltA*, *glyA*, *pgm*, *tkt*, and *uncA*) were amplified by PCR using the primer sets listed in the *Campylobacter* MLST database [12], and the amplified DNA fragments were purified using a FastGene Gel/PCR Extraction Kit (Nippon Genetics Co., Ltd., Tokyo, Japan). Nucleotide sequences of the amplified fragments were determined using the BigDye Terminator 3.1 Cycle Sequencing Kit and 3130-Avant Genetic Analyzer (Applied Biosystems, Waltham, MA, USA). Phylogenetic analyses were conducted using Molecular Evolutionary Genetics Analysis version 11 [13]. The nucleotide sequences of *aspA* (409 base pairs [bp], nucleotide [nt] 669–1077), *glnA* (452 bp, nt 247–698), *gltA* (398 bp, nt 321–719), *glyA* (403 bp, nt 392–794), *tkt* (438 bp, nt 247–684), *uncA* (482 bp, nt 676–1157), and *pgm* (498 bp, nt 421–1118) were used for the phylogenetic analyses.

## 3. Results

### 3.1. Isolation and Identification of Campylobacter spp.

Between 2019 and 2023, we visited 10 dairy farms (9 farms in Songinokhairkhan district were within a 40-km radius of central Ulaanbaatar, while farm 20I in Baganuur district was 130 km away from central Ulaanbaatar), and collected 10 rectal swabs from cattle in each farm. A total of 35 *Campylobacter* spp. were isolated (Table 1), including 23 *C. jejuni* (65.7%) and 4 *C. fetus* (11.4%) identified by multiplex PCR. The remaining isolates were identified as *C. hyointestinalis* (n = 7, 20.0%) by 16S rDNA sequencing analysis and MALDI Biotyper analysis and *C. lari* (n = 1, 2.9%) by MALDI Biotyper analysis. No *C. coli* was isolated in the current study. All *C. jejuni and C. fetus* isolates were also confirmed by MALDI Biotyper analysis. Of the 10 farms, *C. jejuni* was isolated from 9 farms, whereas in farm 19C, only *C. hyointestinalis* was isolated. In farms 19D, 20I, and 23B, only *C. jejuni* was isolated, whereas more than one *Campylobacter* spp. including *C. jejuni*, *C. fetus*, *C. hyointestinalis*, or *C. lari* were isolated from six farms (19A, 19B, 22A, 22B, 23A, and 23C). Two different *Campylobacter* spp., *C. jejuni* and *C. hyointestinalis*, were isolated from cattle ID CA2 in farm 19B and cattle ID CA11 in farm 22B (Table 2).

### 3.2. AMR of Campylobacter spp. Isolates

We previously reported that *C. jejuni* isolates from chickens in Mongolia exhibited extremely high AMR ratios; all the isolates were resistant to TET and half of the isolates were resistant to NA and FQs [9]. Different from the antimicrobial resistance patterns of *C. jejuni* isolates from chickens, most of the *C. jejuni* isolates from cattle in Mongolia were sensitive to TET and CPFX; among the 23 *C. jejuni* isolates, only one isolate (22A-CA10-1) was resistant to CPFX and NA. Nucleotide sequence analysis of *gyrA* of isolate 22A-CA10-1 confirmed that nucleotide substitution at nt257 (ACA to ATA) resulted in a Thr-to-Ile amino acid substitution at codon 86, which is associated with FQ resistance in *C. jejuni* [14]. Consistent with the susceptibility to CPFX, other *C. jejuni* isolates carried Thr at codon 86 (Table 2). *C. hyointestinalis*, *C. fetus*, and *C. lari* are known to be intrinsically resistant to NA [15,16,17], and consistent with this, all three *Campylobacter* spp. isolated in this study were resistant to NA (Table 2).

### 3.3. MLST and Phylogenetic Analysis

For MLST, 17 and 10 *C. jejuni* isolates from cattle (this study) and chickens [9] in Mongolia, respectively, were used. The 17 cattle isolates were classified into 7 STs (ST19, ST22, ST61, ST918, ST2100, ST2217, and ST3098) (Table 2), suggesting the diversity of *C. jejuni* in cattle in Mongolia. Twelve *C. jejuni* isolates from cattle with ST19, ST22, ST61, ST918, and ST2100 belonged to CC19, CC21, CC22, CC48, and CC52, respectively. The CCs of the remaining 5 isolates with ST2217 or ST3098 were not identified. The chicken isolates 15M-B7 and 15M-B18 were assigned to ST14448 (CC21), a new ST (Figure 1).

Figure 1 shows phylogenetic relationships of the isolates from cattle and chickens based on the seven genes used for MLST analysis. Clusters I and III comprised isolates from chickens and cattle, whereas clusters II and IV solely contained cattle isolates. Within cluster II, ST3098 and ST2217 have only been reported in cattle or beef, with the exception of a single human blood isolate, suggesting the association of these STs with cattle [12]. Cluster III included ST22 isolates from both chickens and cattle; however, the cattle isolates were susceptible to TET and CPFX but chicken isolates were resistant to these antibiotics. Cluster IV only included ST61, which is known as a cattle/sheep-associated ST worldwide [8]. ST61 *C. jejuni* were isolated from different farms, suggesting a wide distribution among cattle in Mongolia.

## 4. Discussion

AMR monitoring of *C. jejuni* in humans and food-producing animals conducted by member states of the European Unions revealed extremely high ratios of TET and CPFX resistance in three animal species with sufficient isolate numbers including broilers (ranging from 24.1 to 84.6% and 38.0 to 97.3%, respectively), fattening turkeys (18.8–76.3% and 34.1–93.2%, respectively), and cattle under 1-year old (35.3–92.6% and 45.7–79.3%, respectively), across all nations excluding Nordic countries [18]. Increased trends of CPFX-resistant *C. jejuni* both in human and livestock have been reported in European countries and globally [19,20]. Because EM and CPFX are assigned as critically important antimicrobials for treating severe campylobacteriosis in humans [21], the presence of macrolide- and/or FQ-resistant *Campylobacter* spp. in food-producing animals, foods, and environments represents a public health concern. We previously reported that *C. jejuni* isolated from chickens in Mongolia showed extremely high AMR ratios; all isolates were resistant to TET and nearly half were resistant to FQs [9]. However, this trend was not observed in *C.jejuni* isolates from cattle in this study, as all isolates were susceptible to TET and CPFX, excluding one isolate with CPFX-resistance (22A-CA10-1) carried a Thr-to-Ile substitution at codon 86 in *gyrA*. This unique difference in TET and FQ resistance between chickens and cattle suggests that the transmission of *Campylobacter* spp. between the two livestock species is rare in Mongolia. In the questionnaire survey, we could not obtain any evidence of FQ use on chicken and dairy farms. It has been noted that no fitness burden is required for FQ-resistant *C. jejuni* to colonize the chicken intestine [22,23]. Similarly, it is reported that FQ-resistant *C. jejuni* might have a fitness advantage over FQ-sensitive isolates in colonizing the cattle intestine [24,25]. Thus, although the current prevalence of FQ-resistant *C jejuni* is low in cattle, AMR monitoring, including that of *Campylobacter* spp. in cattle, should be strengthened to detect trends as quickly as possible.

ST61 *C. jejuni* is known to be associated with cattle and sheep [8] and associated with CPFX sensitivity significantly [26]. Consistent with these findings, ST61 was only found in cattle isolates, all of which were sensitive to CPFX. Although undercooked beef is not consumed often because of the Mongolian food culture, *C. jejuni* in cattle can spread to humans via cross-contaminated foods during processing and along the food chain, surface water contaminated with cattle feces, and the consumption of raw milk and dairy products. Indeed, *Campylobacter* spp. are considered as one of the major hazards associated with raw drinking milk [27]. In Japan, where undercooked poultry is the main attribution source of food-borne campylobacteriosis, ST61 *C. jejuni* caused an outbreak by consumption of raw cow milk [28]. Raw ruminant milk is a potential risk of *Campylobacter* infection in humans in Mongolia because it is used for a variety of traditional dairy products such as cream and cheese.

One of the limitations in this study is that most of the samples were collected in the Songinokhairkhan district of Ulaanbaatar, a region where many dairy farms reside. Genetic diversity was observed among the C. jejuni isolates from this district; however, continued analysis of a larger number of Campylobacter spp. isolated from different regions will be required to clarify the characteristics of Campylobacter spp. in cattle in Mongolia.

Although there are no official statistics on campylobacteriosis in humans in Mongolia, a genetic fragment of *Campylobacter* spp. has been detected by PCR in human diarrhea samples of unknown cause (personal communication, Dr. Tundev Odgerel at National Center for Communicable Disease, Mongolia). At present, no information on the attribution source of human campylobacteriosis is available. Future molecular genetic studies of isolates from humans, animals, foods, and environments under the One Health concept will uncover the exposure route of *Campylobacter* spp. to humans.

## 5. Conclusions

We recently reported the high antimicrobial resistance ratios of *C. jejuni* in chickens in Mongolia; all isolates were resistant to TET, and nearly half were resistant to FQs. However, this study revealed that the antimicrobial resistance patterns of the *C. jejuni* cattle isolates completely differed from those of chicken isolates. This is the first report on the characterization of *Campylobacter* spp. in cattle in Mongolia. Data on *Campylobacter* spp. in food-producing animals will be valuable for investigating potential sources and infection routes to humans.

## Figures and Tables

**Figure 1 vetsci-12-00931-f001:**
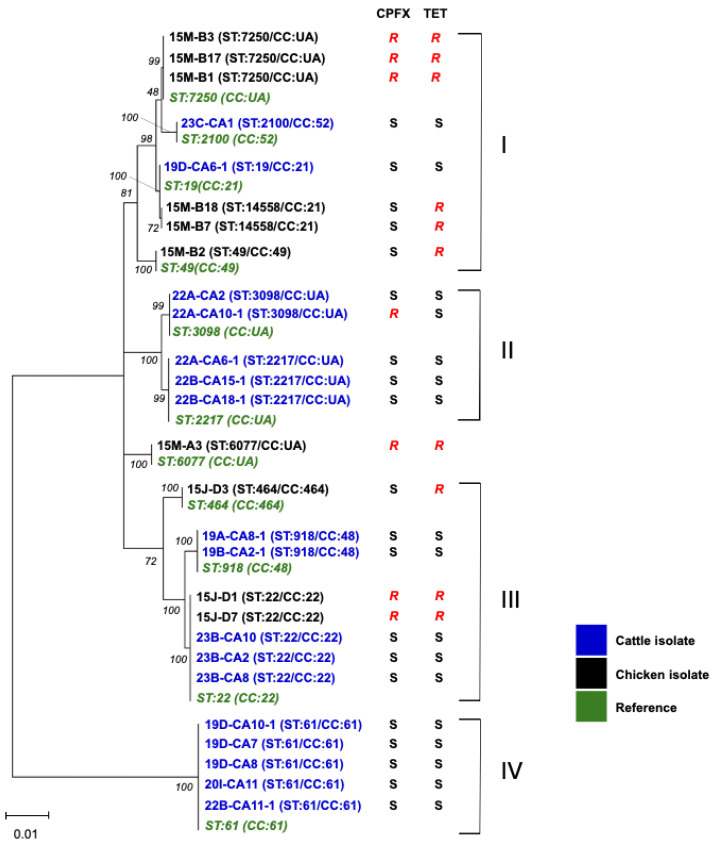
Phylogenetic analysis of *C. jejuni* isolates from cattle and chickens in Mongolia. Nucleotide sequences of a total of 3080 nt from *aspA*, *glnA*, *gltA*, *glyA*, *tkt*, *uncA*, and *pgm* were used. The nucleotide sequences of ST19 (CC21), ST22 (CC22), ST49 (CC49), ST464 (CC464), ST918 (CC48), ST2100 (CC52), ST2217 (CC:UA), ST3098 (CC:UA), ST6077 (CC:UA), ST7250 (CC:UA), ST14558 (CC21), and ST61 (CC61) were obtained from the *Campylobacter* Sequence Typing database [12]. The percentages (50%) of replicate trees in which the associated taxa clustered together in the bootstrap test (1000 replicates) are shown in italic numbers next to the branches.

**Table 1 vetsci-12-00931-t001:** *Campylobacter* spp. isolated from cattle.

Location (District of Ulaanbaatar)	Sampling Year	Farm ID	Number of Animals ^1^	Number of Isolates	Number of *Campylobacter* spp. Isolated			
					*C. jejuni*	*C. fetus*	*C. lari*	*C. hyointestinalis*
Songinokhairkhan	2019	19A	10	3	1	1	- ^2^	1
Songinokhairkhan	2019	19B	10	4	2	1	-	1
Songinokhairkhan	2019	19C	10	3	-	-	-	3
Songinokhairkhan	2019	19D	10	4	4	-	-	-
Baganuur	2020	20I	10	1	1	-	-	-
Songinokhairkhan	2022	22A	10	4	3	-	1	-
Songinokhairkhan	2022	22B	10	5	3	-	-	2
Songinokhairkhan	2023	23A	10	2	1	1	-	-
Songinokhairkhan	2023	23B	10	5	5	-	-	-
Songinokhairkhan	2023	23C	10	4	3	1	-	-
		Total	100	35	23	4	1	7

^1^ Number of cattle from which rectal swabs were collected. 2 Not isolated.

**Table 2 vetsci-12-00931-t002:** ST and antimicrobial resistance of *Campylobacter* isolates.

Sample ID ^(1)^	*Campylobacter* spp.	ST ^(2)^	CC ^(2)^	MIC (μg/mL) ^(3)^				*gyrA* Codon 86
				TET	EM	CPFX	NA	
19A-CA8-1	*C. jejuni*	918	48	0.14	0.5	0.79	1.25	Thr
19B-CA2-1	*C. jejuni*	918	48	0.048	0.63	0.14	1	Thr
19B-CA8	*C. jejuni*	NT ^(4)^	NT	0.26	0.24	0.079	1.5	Thr
19D-CA6-1	*C. jejuni*	19	21	0.0945	0.75	0.157	2	Thr
19D-CA7	*C. jejuni*	61	61	0.237	0.75	0.313	7	Thr
19D-CA8	*C. jejuni*	61	61	0.594	1.5	0.25	8	Thr
19D-CA10-1	*C. jejuni*	61	61	0.444	1	0.765	4.75	Thr
20I-CA11	*C. jejuni*	61	61	0.094	2	0.125	5.5	Thr
22A-CA2	*C. jejuni*	3098	UA ^(4)^	0.074	2.13	0.07	0.75	Thr
22A-CA6-1	*C. jejuni*	2217	UA	0.547	2.13	0.05	0.5	Thr
22A-CA10-1	*C. jejuni*	3098	UA	0.059	0.75	>32	20	Ile
22B-CA11-1	*C. jejuni*	61	61	0.079	1.13	0.172	3	Thr
22B-CA15-1	*C. jejuni*	2217	UA	0.133	0.813	0.01	0.025	Thr
22B-CA18-1	*C. jejuni*	2217	UA	0.137	2.14	0.016	0.5	Thr
23A-CA1	*C. jejuni*	NT	NT	0.035	0.315	0.035	1.75	Thr
23B-CA2	*C. jejuni*	22	22	0.158	1.125	0.125	2	Thr
23B-CA5	*C. jejuni*	NT	NT	0.142	1	0.094	2	Thr
23B-CA8	*C. jejuni*	22	22	0.11	0.75	0.094	2	Thr
23B-CA9	*C. jejuni*	NT	NT	0.095	1	0.064	1.5	Thr
23B-CA10	*C. jejuni*	22	22	0.127	0.875	0.094	2	Thr
23C-CA1	*C. jejuni*	2100	52	0.21	1.3	0.136	4	Thr
23C-CA6	*C. jejuni*	NT	NT	0.158	1.5	0.079	2	Thr
23C-CA8	*C. jejuni*	NT	NT	0.056	0.75	0.158	4	Thr
19A-CA4-1	*C. hyointestinalis*	NT	NT	0.38	1.5	0.19	144	NT
19B-CA2-2	*C. hyointestinalis*	NT	NT	0.38	2	0.19	144	NT
19C-CA3	*C. hyointestinalis*	NT	NT	0.158	1.25	0.25	48	NT
19C-CA7-1	*C. hyointestinalis*	NT	NT	0.064	0.75	3	256	NT
19C-CA9	*C. hyointestinalis*	NT	NT	0.19	0.845	0.25	40	NT
22B-CA11-2	*C. hyointestinalis*	NT	NT	0.094	0.625	0.125	>256	NT
22B-CA13-1	*C. hyointestinalis*	NT	NT	0.095	0.438	0.079	32	NT
19A-CA1-1	*C. fetus*	NT	NT	0.57	1	0.5	160	NT
19B-CA4-1	*C. fetus*	NT	NT	0.314	0.75	0.262	>256	NT
23A-CA7	*C. fetus*	NT	NT	0.315	0.625	0.138	>256	NT
23C-CA7	*C. fetus*	NT	NT	0.274	0.625	0.208	192	NT
22A-CA5-1	*C. lari*	NT	NT	0.158	2.25	5.5	>256	NT

^(1)^ The numbers 19, 20, 22, and 23 indicate year of sampling (2019, 2020, 2022, and 2023), while following single capital letters indicate farm ID. CA with numbers mean cattle ID and numbers after hyphens indicates representative colony ID of the same cattle. ^(2)^ ST: Sequence type; CC: clonal complex of *C. jejuni.*
^(3)^ MIC: minimum inhibitory concentrations. Average of two independent experiments are shown. TET, tetracycline; EM, erythromycin; CPFX, ciprofloxacin; NA, nalidixic acid. MICs for NA was determined using microbroth dilution assay, while those of remaining antibiotics were determined using E-test. ^(4)^ NT: not tested; UA: unassigned.

## Data Availability

The original contributions presented in this study are included in the article. Nucleotide sequences of representative isolates of each ST, ST19: 19D-CA6-1; ST22: 23B-CA10; ST61: 19D-CA10-1; ST918: 19A-CA8-1; ST2100: 23C-CA1; ST2217: 22A-CA6-1; ST-3098: 22A-CA2; ST14558: 15MA-B18, have been deposited to DDBJ. Accession numbers assigned are from LC888096 to LC888151. Further inquiries can be directed to the corresponding author(s).

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
