# Peer review of "Antimicrobial Susceptibility of Campylobacter spp. Isolated from Cattle in Mongolia"

_vetsci, 2025, doi:10.3390/vetsci12100931_

Round 1

Reviewer 1 Report

Comments and Suggestions for Authors

The author claims that this is the first report on the characteristics of the genus Campylobacter in cattle in Mongolia. Although there are currently no official statistics on human Campylobacter disease in Mongolia, but I believe this article has the following main issues:

  • The main purpose of the author is to evaluate the prevalence and antibiotic resistance of Campylobacter jejuni. The antibiotic resistance pattern of the isolated bovine isolates is completely different from the previously reported chicken isolates. The purpose and significance of this analysis are mainly discussed?
  • The collection of samples from cows in the surrounding dairy farms of Ulaanbaatar city revealed rich genetic diversity through the multi focal sequence typing of the isolated bovine isolates of Campylobacter jejuni. I believe that such sampling does not match the conclusions, and the sampling is one-sided and has serious limitations.
  • Table 2 requires a strict three line chart, and Figure 1 is not clear enough.

Author Response

Thank you very much for your careful reading of our manuscript. Followings are our answers and changes that we have made in response to the concerns raised by the reviewer #1.

Comment 1: The main purpose of the author is to evaluate the prevalence and antibiotic resistance of Campylobacter jejuni. The antibiotic resistance pattern of the isolated bovine isolates is completely different from the previously reported chicken isolates. The purpose and significance of this analysis are mainly discussed?

Response 1: In developed countries, poultry is the primary source of foodborne Campylobacter infection and ruminants are considered as the secondary source. This may also apply to Mongolia, where Westernization of food culture and the spread of ready-to-eat foods are gradually progressing. However, Mongolia possesses a unique culture of utilizing animal milk, potentially posing a high risk of Campylobacter infections originating from ruminants. This study was conducted because no reports are available on the distribution of Campylobacter spp. in ruminants or their antimicrobial resistance. We have described these backgrounds and purpose in Introduction (in line 57-62 and 70-77 of the original manuscript; in line 68-73 and 85-93 of the revised manuscript). However, to emphasize the significance of AMR analysis of Campylobacter spp. isolates, we have added the following description to lines 77-80 of the revised manuscript.

“Because WHO assigned EM and CPFX as critically important antimicrobials for treating severe campylobacteriosis in humans, the presence of macrolide- and/or FQ-resistant Campylobacter spp. in food-producing animals, foods, and environments represents a public health concern.”

Comment 2: The collection of samples from cows in the surrounding dairy farms of Ulaanbaatar city revealed rich genetic diversity through the multi focal sequence typing of the isolated bovine isolates of Campylobacter jejuni. I believe that such sampling does not match the conclusions, and the sampling is one-sided and has serious limitations.

Response 2: We agreed to the reviewer’s comment and have added the limitation of study as follows (in line 252-256 of the revised manuscript).

“One of the limitations in this study is that most of the samples were collected in the Songinokhairkhan district of Ulaanbaatar, a region where many dairy farms reside. Genetic diversity was observed among the C. jejuni isolates from this district; however, continued analysis of a larger number of Campylobacter spp. isolated from different regions will be required to clarify the characteristics of Campylobacter spp. in cattle in Mongolia”

Comment 3: Table 2 requires a strict three line chart, and Figure 1 is not clear enough.

Response 3: Thank you for the comment. We have modified Figure 1 as follows: 1) The topmost cluster was misaligned with its phylogenetic tree position. We have corrected it in the revised manuscript. 2) We have changed the order of the isolates in each branch, sequences of representative ST that were obtained from database were indicated at the bottom. 3) Resistance to TET or CPFX were indicated with italic. We have also modified the Table 2 as follows: 1) We have changed the order of isolates by species, 2) Table 2 was formatted onto a single page.

Reviewer 2 Report

Comments and Suggestions for Authors

This is a timely and important study that investigates the antimicrobial susceptibility of Campylobacter spp. isolated from cattle in Mongolia, a region traditionally considered to have low levels of antimicrobial resistance. The emergence of resistant strains in a region where surveillance data is scarce is of significant scientific interest. The methodology is sound, the results are robust, and the comparison between isolates from cattle and chickens is a particularly compelling aspect of the study.

While the molecular epidemiological analysis is limited to MLST and QRDR, this paper provides valuable baseline data on the antimicrobial resistance landscape in Mongolia. Therefore, I believe this work is worthy of publication in the journal, perhaps as a Brief Report.

To further enhance the scientific presentation of this valuable work, I offer the following suggestions as a fellow scientist.

Specific Comments

Readability of Table 2: While Table 2 is comprehensive, its current format makes it difficult for readers to interpret the data at a glance.

The "NT" (Not Tested) notation is visually dominant and tends to obscure the overall trends in the data.

The table is split across two pages, which impairs readability. I recommend formatting the table to fit onto a single page.

Please consider reorganizing the data by species rather than by year. Unless there is a specific year-over-year trend that is critical to the manuscript's conclusions, organizing by species would likely provide a clearer and more direct comparison.

Formatting of Figure 1: The figure itself is clear and well-presented; however, some minor formatting changes are needed.

The caption is currently embedded within the image. I suggest moving this to a separate figure legend, as is standard journal practice.

The figure title and the description on line 105 appear to be redundant. Please revise this for conciseness.

Thank you for your valuable contribution to the surveillance of antimicrobial resistance in Mongolia.

Author Response

Thank you very much for your careful reading of our manuscript. Followings are our answers and changes that we have made in response to the concerns raised by the reviewer #2.

Comments 1-4: 

  1. Readability of Table 2: While Table 2 is comprehensive, its current format makes it difficult for readers to interpret the data at a glance.
  2. The "NT" (Not Tested) notation is visually dominant and tends to obscure the overall trends in the data.
  3. The table is split across two pages, which impairs readability. I recommend formatting the table to fit onto a single page.
  4. Please consider reorganizing the data by species rather than by year. Unless there is a specific year-over-year trend that is critical to the manuscript's conclusions, organizing by species would likely provide a clearer and more direct comparison.

Response 1-4: Thank you for the reviewer’s suggestions. According to the reviewer’s suggestions, we have changed the order of isolates by species and formatted Table 2 onto a single page. This may also clarify the reviewer’s concern on “NT”, since it shows more clearly that MLST and gyrA substitution were analyzed for C. jejuni isolates in this study.

Comment 5: Formatting of Figure 1: The figure itself is clear and well-presented; however, some minor formatting changes are needed.

Response 5: The topmost cluster was misaligned with its phylogenetic tree position, thus we have corrected it in the revised manuscript. We have changed the order of the isolates in each branch, sequences of representative ST that were obtained from database were indicated at the bottom. Resistance to TET or CPFX were indicated with italic.

Comment 6: The caption is currently embedded within the image. I suggest moving this to a separate figure legend, as is standard journal practice.

Response 6: As pointed out by the reviewer, we have removed duplicated figure legend.

Comment 7: The figure title and the description on line 105 appear to be redundant. Please revise this for conciseness.

Response 7: We have removed “for phylogenetic analysis” (in lines 106-107 of the original manuscript) and “for C. jejuni” (in line 109 of the original manuscript) from the legend for Figure 1.

Reviewer 3 Report

Comments and Suggestions for Authors

Line 20: not only are cattle the main reservoir of C. for humans, but also poultry

Line 68: correct the spelling of the word "Additionalany"

It is unclear how the typing was performed. Was C. hyointestinalis analysed using MALDI? Because in line 57, it is written "by sequencing" as well for C. lari

Figure 1 has two identical legends, please discard one of them

Author Response

Thank you very much for your careful reading of our manuscript. Followings are our answers and changes that we have made in response to the concerns raised by the reviewer #3.

Comment 1: Line 20: not only are cattle the main reservoir of C. for humans, but also poultry.

Response 1: Thank you for the reviewer’s comments. We have changed this from “Cattle” to “Poultry and cattle” (in line 30 in the revised manuscript)

Comment 2: Line 68: correct the spelling of the word "Additionalany"

Response 2: We thank the reviewer for pointing out the typographical error. We have corrected this to “Additionally” (line 82 of the revised manuscript)

Comment 3: It is unclear how the typing was performed. Was C. hyointestinalis analysed using MALDI? Because in line 57, it is written "by sequencing" as well for C. lari

Response 3: We thank the reviewer’s comments. We have changed the sentences as follows to clarify the reviewer’s question. “The remaining isolates were identified as C. hyointestinalis (n=7, 20.0%) by 16S rDNA sequencing analysis and MALDI Biotyper analysis and C. lari (n=1, 2.9%) by MALDI Biotyper analysis. No C. coli was isolated in the current study. All C. jejuni and C. fetus isolates were also confirmed by MALDI Biotyper analysis.” (in line 159-161 of the revised manuscript)

Comment 4: Figure 1 has two identical legends, please discard one of them

Response 4: We have corrected this duplication in the revised manuscript.

Round 2

Reviewer 1 Report

Comments and Suggestions for Authors

I think the quality of the revised manuscript has improved, but my previous important concerns have not been effectively addressed. I have to adhere to my strict review standards.